# Toxicity of Different Types of Surfactants via Cellular and Enzymatic Assay Systems

**DOI:** 10.3390/ijms24010515

**Published:** 2022-12-28

**Authors:** Oleg S. Sutormin, Elizaveta M. Kolosova, Irina G. Torgashina, Valentina A. Kratasyuk, Nadezhda S. Kudryasheva, Julia S. Kinstler, Devard I. Stom

**Affiliations:** 1Department of Biophysics, Institute of Fundamental Biology and Biotechnology, Siberian Federal University, 660041 Krasnoyarsk, Russia; 2Photobiology Laboratory, Institute of Biophysics, Federal Research Center ‘Krasnoyarsk Science Center, Siberian Branch of the Russian Academy of Sciences’, 660036 Krasnoyarsk, Russia; 3Department of Vertebrate Zoology and Ecology, Faculty of Biology and Soil, Irkutsk State University, 664003 Irkutsk, Russia; 4Baikal Museum of the Siberian Branch of the Russian Academy of Sciences, 664520 Irkutsk, Russia; 5Department of Civil and Environmental Engineering, School of Architecture, Construction and Design, Irkutsk National Research Technical University, 664074 Irkutsk, Russia

**Keywords:** surfactants, enzyme-inhibition-based assay, bioluminescent assay, luminous bacteria, ecotoxicity

## Abstract

Surfactants have a widespread occurrence, not only as household detergents, but also in their application in industry and medicine. There are numerous bioassays for assessing surfactant toxicity, but investigations of their impact on biological systems at the molecular level are still needed. In this paper, luminous marine bacteria and their coupled NAD(P)H:FMN-oxidoreductase + luciferase (Red + Luc) enzyme system was applied to examine the effects of different types of surfactants, including cationic cetyltrimethylammonium bromide (CTAB), non-ionic polyoxyethylene 20 sorbitan monooleate (Tween 80) and anionic sodium lauryl sulfate (SLS), and to assess whether the Red + Luc enzyme system can be used as a more sensitive indicator of toxicity. It was shown that the greatest inhibitory effect of the surfactants on the activity of luminous bacteria and the Red + Luc enzyme system was in the presence of SLS samples. The calculated IC_50_ and EC_50_ values of SLS were 10^−5^ M and 10^−2^ M for the enzymatic and cellular assay systems, respectively. The results highlight the benefits of using the enzymatic assay system in ecotoxicology as a tool for revealing surfactant effects on intracellular proteins if the cellular membrane is damaged under a long-term exposure period in the presence of the surfactants. For this purpose, the bioluminescent enzyme-inhibition-based assay could be used as an advanced research tool for the evaluation of surfactant toxicity at the molecular level of living organisms due to its technical simplicity and rapid response time.

## 1. Introduction

Surfactants have a widespread occurrence, not only as household detergents, but also in their application in industry and medicine. Previously, amphiphilic surfactants have been shown to have a positive impact on the adsorption of pollutants from soil and water samples [1]. Amphiphilic surfactants can also be used in drug production [2]. On the other hand, surfactants are predominant water pollutants due to urban and municipal wastewater discharges [3]. There is no doubt that the presence of surfactants in the natural environment might lead to toxicity effects of the surfactants on the cellular or molecular level of the organization of living things. For instance, surfactants are carcinogenic agents, and they show high chronic and sublethal toxicity effects on aquatic organisms, usually at concentrations from 0.4 to 40 mg/L [4]. However, when ciliates *Tetrahymena pyriformis* were used as test organisms, they showed a 50% lethal dose (LD_50_) for surfactants at concentrations from 0.09 mg/L [5]. Furthermore, the toxicity features of surfactants depend on their hydrocarbon chain length and degree of linearity [6]. Basically, the chemical structure of surfactants correlates with their toxicity level [7]. For example, anionic surfactants are more toxic than non-ionic surface-active agents [8]. However, antagonistic interaction was shown under the joint action of different surfactants [9].

There are numerous articles describing the results of the assessment of toxic effects of surfactants using biotesting methods [10,11,12,13]. Micro- and macroalgae, seed plants, invertebrates, fish and bacteria are frequently used as test objects in toxicity bioassays [6,14]. Summarizing the published data, the following conclusion could be made: the luminous bacteria *Vibrio fischeri* and *Photobacterium phosphoreum* have the greatest sensitivity and rapid response time to the toxic effects of surfactants in wastewater among other bioassays [8,11,15]. This result raises the question as to how surfactants affect the molecular level of luminous bacteria, resulting in their great sensitivity to these substances. To answer this question, investigating the sensitivity of enzymes from the luminous bacteria is a desired acceptable option. Here, a bacterial coupled-enzyme system, which involves two enzymes, namely, NAD(P)H:FMN-oxidoreductase and luciferase (Red + Luc), is usually used as a convenient and rapid tool to estimate toxicity in environmental monitoring [16]. The principle of bioluminescent enzymatic bioassays is to identify toxic properties of chemicals and mixtures based on their influence on the parameters of bioluminescent coupled-enzymatic reactions [17]. The bioluminescent coupled-enzyme system was previously shown to have sensitivity and specificity to substances, such as heavy metals, pharmaceuticals, quinones, etc. [16,18,19,20]. Moreover, the Red + Luc enzyme system was employed for the environmental bioassay of soil systems [21,22,23,24,25,26].

Therefore, this paper aims to examine and compare the effects of different types of surfactants, including cetyltrimethylammonium bromide (CTAB), polyoxyethylene 20 sorbitan monooleate (Tween 80) and sodium lauryl sulfate (SLS), on bioluminescent bacteria cells and the Red + Luc enzyme system for assessing whether the Red + Luc enzyme system can be used as a more sensitive indicator of toxicity.

## 2. Results

### 2.1. The Effect of Surfactants on the Bioluminescent Coupled Red + Luc Enzyme System

The effect of commercial surfactants of different types (Tween 80, CTAB, SLS) on the coupled-enzyme system Red + Luc was determined (Figure 1). CTAB, at concentrations from 10^−9^ M to its critical concentration of micelle formation (CCM), did not produce any impact on the activity of the Red + Luc enzyme system. In the same manner as CTAB, Tween 80 concentrations, which were lower than the Tween 80 CCM concentration, did not influence the activity of the coupled-enzyme system (Figure 1b). An opposite result was obtained for the SLS inhibitory effect on the Red + Luc enzyme system. The SLS samples decreased the activity of the coupled-enzyme system Red + Luc more effectively than the other surfactant samples. The concentration of SLS equal to 10^−5^ M led to an approximately 55% lowering of the Red + Luc enzymatic activity. Additionally, the SLS concentration above and below its CCM showed almost the total inhibition of the Red + Luc enzyme system activity.

The half-maximum inhibitory concentrations (IC_50_) for SLS were calculated (Table 1).

### 2.2. The Effect of the Surfactants on the Intact Marine Luminous Bacterium Strain P. phosphoreum

The effect of commercial surfactants (CTAB, Tween 80, SLS) on the bioluminescent bacteria *P. phosphoreum* was determined (Figure 2). The results demonstrate that the marine luminous bacteria did not show any valuable sensitivity to the presence of the CTAB and Tween 80 samples. The luminescence intensity of the bacteria did not change in the presence of the surfactants. On the other hand, the presence of the SLS samples affected the luminescence intensity of the bacterium strain *P. phosphoreum*. The higher the concentration of SLS added, the lower the intensity of luminous bacterium luminescence measured because among the three different types of surfactants investigated, the bioluminescent bacteria *P. phosphoreum* was found to be the most sensitive to the effect of the SLS samples. Additionally, it should be noted that the concentrations equal to or above the CCM surfactant concentrations did not lead to a crucial reduction in the luminescence intensity of the bacteria. As for the coupled bioluminescent enzyme system, the median effective concentration value (EC_50_) for SLS was calculated (Table 1).

## 3. Discussion

In the present paper, three types of surfactants (anionic, non-ionic and cationic) were studied. As a result, the surfactants were found to inhibit the activities of the Red + Luc enzyme system more than the light intensities of the luminous bacteria. The low activity of the cellular and enzyme-based assays in the presence of the surfactants at concentrations above CCM is attributed to the aggregation of the surfactant molecules into micelles [27,28]. This surfactant aggregation could cause oxygen diffusion limitation, which has negative effects on the process of the bioluminescence reaction for both the Red + Luc enzyme system and the luminous bacteria [16,29]. In this case, we mainly focused our attention on the speculation of the impact of the surfactants on the activities of the enzymatic assay system at concentrations that are lower than CCM. Firstly, these concentrations are more relevant and could help to understand how the chemical structure of the surfactant correlates with its toxicity effects. Secondly, surfactants at the CCM concentration and higher form a heterogeneous medium which is difficult to dissolve in water [27]. The formation of this medium results in the viscosity values of the reaction mixture in the presence of the surfactants, which could negatively affect the Red + Luc enzymatic activity. It was previously shown that the viscosity values of the reaction mixture above 3 cP negatively affected the activity of the Red + Luc enzyme system [30].

Comparing the sensitivity of two assay systems of one organism showed that both assay systems have a sensitivity to the anionic surfactant, presented by SLS. IC_50_ of the coupled-enzyme system for the surfactant was calculated to be 10^−5^ M and EC_50_ of the marine bacteria for SLS was equal to 10^−2^ M. Additionally, the ecological relevance for environmental toxicology implies an SLS concentrations from 0.5 to 4 mg/L because these concentrations are found in natural waters [31]. The data obtained in this study showed that SLS at these concentrations, from 0.5 to 4 mg/L, inhibited the activity of the Red + Luc enzyme system from 80% to 42%, which could be classified as a toxic effect. However, there were no significant effects when the luminous bacteria were exposed to this chemical (Figure 2c). It seems that the obtained toxicity of SLS at the above-mentioned concentrations is still a matter of debate. On the one hand, such toxicity will not be observed under natural conditions, which is due to the fact that enzymes are not exposed to the surfactants as they were in the experimental conditions of the present study. On the other hand, there is a need for additional investigations regarding the monitoring of the glowing activity of luminous bacteria under the exposure to long-term SLS concentrations, from 0.5 to 4 mg/L. Alternatively, cationic and non-ionic surfactants, CTAB and Tween 80, respectively, did not show any negative effects on the light intensity values of the marine luminous bacteria (Figure 2a,b) and the activity of the Red + Luc enzyme system (Figure 1a,b) at surfactant concentrations lower than CCM. Furthermore, a decrease in the concentrations of the surfactants to or below CMM in the coupled-enzyme system is probably associated with micelle formation and a change in the viscosity values of the reaction mixture. In case of the luminous bacteria, the same lowering of the light intensity values at concentrations of CTAB above CCM is connected with the surfactant antibacterial properties at concentrations above CCM [27]. Additionally, the absence of toxicity effects of Tween 80 on the luminous bacterium *P. phosphoreum* and the Red + Luc enzyme system might be a result that confirms the fact that the application of Tween 80 in bioremediation approaches [32] is environmentally friendly.

Moreover, the sensitivity of the applied assay systems was compared with other aquatic species exposed to these surfactants. The results are presented in Table 2. The data presented in Table 2 show that *P. phosphoreum* is not always a test system with a higher sensitivity to the surfactants. For instance, the guppy fish *Poecilia reticulata* has a greater sensitivity to SLS. Such a sensitivity is likely to be connected with the ability of anionic SLS to create a film on the water surface, limiting oxygen supply into the water system and altering its dissolution [33]. In addition, the luminescent bacteria involved in this study had sensitivity to SLS, which agrees with previously published results [34]. At the same time, it seems that there is a lack of data regarding the investigations of the Tween 80 toxicity. As was mentioned earlier, Tween 80 is used in surfactant-enhanced technology for the remediation of soils contaminated with hydrophobic organic compounds [35,36]. Therefore, it is quite difficult to find results regarding chronic and sublethal direct-toxicity effects of Tween 80 on any organisms. In this case, the results obtained in this paper regarding the toxicity effects of Tween 80 on the bioluminescent bacteria and the Red + Luc enzyme system might be used as an additional supporting point for approving the perspective soil remediation by the technology involving Tween 80.

As one can see, Table 2 presents only the cellular assay system for the assessment of the toxicity effects of the surfactants, rather than the enzymatic ones. Though, as shown in this study, the Red + Luc enzyme system has a greater sensitivity to SLS than the cellular one. The high sensitivity of the enzymatic assay systems is connected with the fact that enzymatic systems do not have the protection mechanisms that cellular systems have, such as a cellular membrane. Thus, the toxicity assessment of chemicals based on enzymatic assay systems could provide knowledge regarding the direct impact of chemicals on the most important metabolic pathways in an organism. Additionally, the enzymatic assay systems are free from the conditions when the toxicant has a positive effect on the informative parameters of the cellular assay system. For instance, it was reported that CTAB at concentrations of 1 × 10^−10^ and 1 × 10^−11^ mol/L had a stimulating effect on the activity of Gram-positive bacterium *Bacillus subtilis 6633* [38].

## 4. Materials and Methods

### 4.1. Surfactants

Three different types of commercial surfactants were evaluated: cationic, CTAB (PanReac Applichem, Chicago, IL, USA); non-ionic, Tween 80 (PanReac Applichem, Chicago, IL, USA); and anionic, SLS (PanReac Applichem, Chicago, IL, USA). The surface-active properties of each surfactant are described in Table 3.

### 4.2. Bioluminescent Enzyme-Inhibition-Based Assay

The bioluminescent enzymatic bioassay was carried out using lyophilized preparations of highly purified enzymes, produced in the laboratory of Nanobiotechnology and Bioluminescence at the Institute of Biophysics SB RAS (Krasnoyarsk, Russia). Each vial of the lyophilized preparation of the enzymes contained 0.4 mg/mL luciferase EC 1.14.14.3 from the recombinant strain *E. coli* and 0.18 units of NAD(P)H:FMN-oxidoreductase EC 1.5.1.29 from the *Vibrio fischeri* culture collection IBSO 836. To prepare enzyme solutions, 5 mL of a 0.05 M potassium phosphate buffer (pH 6.9) was added to a vial containing the enzymes. FMN (Serva, Heidelberg, Germany), NADH (Sigma, Steinheim, Germany) and tetradecanal (Merck, Steinheim, Germany) were used as substrates of the Red + Luc system. A 0.0025% (*v*/*v*) solution of myristic aldehyde was prepared by mixing 50 µL of 0.25% (*v*/*v*) ethanol solution of aldehyde and 5 mL of 0.05 M potassium phosphate buffer (pH 6.9). The NADH solution was prepared in a 0.05 M potassium phosphate buffer (pH 6.9). The activity of the coupled Red + Luc enzyme system was measured in the reaction mixture containing 300 µL of a 0.05 M potassium phosphate buffer (pH 6.9), 5 µL of an enzyme solution, 50 µL of a 0.0025% (*v*/*v*) aldehyde solution, 100 µL of a 0.4 mM NADH solution and 10 µL of a 0.5 mM FMN solution. In the beginning, the control luminescence intensity of the enzyme system (I_0_) was registered. To identify I_0_, all the components of the reaction mixture and 50 µL of distilled water (control solution) were subsequently added to the tube of a Glomax 20/20^n^ luminometer (Promega, Sunnyvale, CA, USA) and quickly mixed, and the maximum luminescence intensity was measured. To determine the luminescence intensity in the presence of the surfactant solution (I), 50 µL of the control solution was placed into 50 µL of the investigated surfactant solution. The time of the bioluminescent analysis was 300 s. All the measurements were repeated 3 times with a deviation of no more than 10%. The residual luminescence value (I/I_0_, %) was used as an indicator of the total toxicity of the soil samples. The residual luminescence value was calculated according to the formula (I/I_0_) × 100%. With I/I_0_ > 80%, the analyzed surfactant sample was considered non-toxic. With 50% < I/I_0_ < 80%, the sample was considered toxic, and with I/I_0_ < 50%, the sample being studied was considered highly toxic. The investigated concentrations of the surfactant solutions were the following: CTAB, 10^−2^–10^−10^ M; Tween 80, 10^−2^–10^−7^ M; SLS, 10^−1^–10^−9^ M. The value of the inhibition parameter IC_50_ (concentrations of the surfactants inhibiting the enzyme system by 50%) were determined.

### 4.3. Luminous Bacteria Assay

An intact marine luminous bacterium strain *P. phosphoreum* 1883 IBSO [41] was used to evaluate the effects of surfactants on the cellular system. The strain was obtained from the Collection of Luminous Bacteria CCIBSO-863, Institute of Biophysics SB RAS, Krasnoyarsk, Russia. For the cultivation of *P. phosphoreum*, the semisynthetic medium containing 10 g/L Tryptone, 28.5 g/L NaCl, 4.5 g/L MgCl_2_·6H_2_O, 0.5 g/L CaCl_2_, 0.5 g/L KCl, 3 g/L yeast extract and 12.5 g/L Agar was used. NaCl was of analytical grade, and it was applied to prepare bacterial suspensions for the bioluminescence measurements. Solutions of 3% NaCl were used to imitate a marine environment for the bacterial cells and to balance osmotic processes.

The light intensity of the intact marine luminous bacteria was measured in the reaction mixture containing 450 µL bacterial suspension and 50 µL distilled water or 50 µL surfactant solution. In the beginning, the control luminescence intensity of the cellular system (I_0_) was registered. To identify I_0_, 450 µL of the bacterial suspension and 50 µL of distilled water (control solution) were subsequently added to the tube of a Glomax 20/20^n^ luminometer (Promega, Sunnyvale, CA, USA) and quickly mixed, and the maximum luminescence intensity was measured. To determine the luminescence intensity in the presence of the surfactants (I), 50 µL of the control solution was placed into 50 µL of the investigated surfactant solution. The time of the bioluminescent analysis was 60 s. All the measurements were repeated 5 times with a deviation of no more than 20%. The residual luminescence value (I/I_0_, %) was used as an indicator of the total toxicity of the surfactant samples. The residual luminescence value was calculated according to the following formula: (I/I_0_) × 100%. With I/I_0_ > 80%, the analyzed surfactant was considered non-toxic. With 50% < I/I_0_ < 80% the sample was considered toxic, and with I/I_0_ < 50%, the sample being studied was considered highly toxic. The surfactant concentrations were analyzed in the following ranges for CTAB, Tween 80 and SLS, respectively: from 10^−2^ to 10^−10^ M; from 10^−2^ to 10^−7^ M; and from 10^−1^ to 10^−9^ M. The half-maximum effective concentrations EC_50_ (concentrations of surfactants inhibiting the bioluminescent bacteria by 50%) were determined.

### 4.4. Data Processing

The data are presented as a mean value (M) ± standard deviation (s). All the measurements were repeated 3 to 5 times. The significance of differences was determined by Student’s *t*-test. The results were considered statistically significant at *p* < 0.05.

## 5. Conclusions

In the present paper, the toxicity effects of different types of surfactants on cellular, intact marine luminous bacteria and enzymatic and coupled Red + Luc enzyme assay systems were studied. The applied assay systems showed their sensitivity to the anionic surfactant presented by SLS. The calculated values of the parameters IC_50_ and EC_50_ for the Red + Luc enzyme system and luminous bacteria were 10^−5^ M and 10^−2^ M, respectively. Additionally, the absence of negative effects of Tween 80 on the cellular and enzymatic assay systems could be considered as supporting information for the safe use of Tween 80 surfactant-enhanced technology for the remediation of soils contaminated with different toxicants. Meanwhile, little is known about how readily surfactants might affect intracellular proteins if the cellular membrane is damaged under a long-term exposure period. In this case, the bioluminescent enzymatic method is suitable for revealing the potential toxicity of surfactants at the molecular level of organization of living systems. It could be used as a basis for new methods of screening various surfactants. The bioluminescent enzyme-inhibition-based assay is simple, takes only 2–3 min and has a higher sensitivity than other bioassays in toxicology methods.

## Figures and Tables

**Figure 1 ijms-24-00515-f001:**
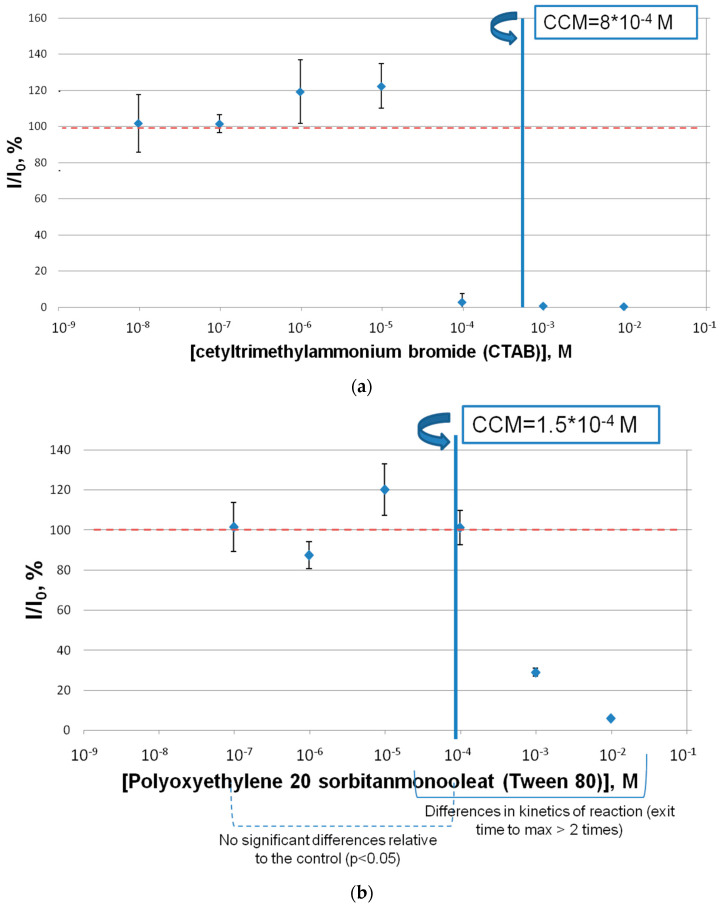
The residual luminescence intensity, I/I_0_, of the coupled-enzyme system Red + Luc in the presence of Tween 80 (**a**), CTAB (**b**) and SLS (**c**). The red dotted line means the activity of the Red + Luc enzyme system in the absence of the surfactants.

**Figure 2 ijms-24-00515-f002:**
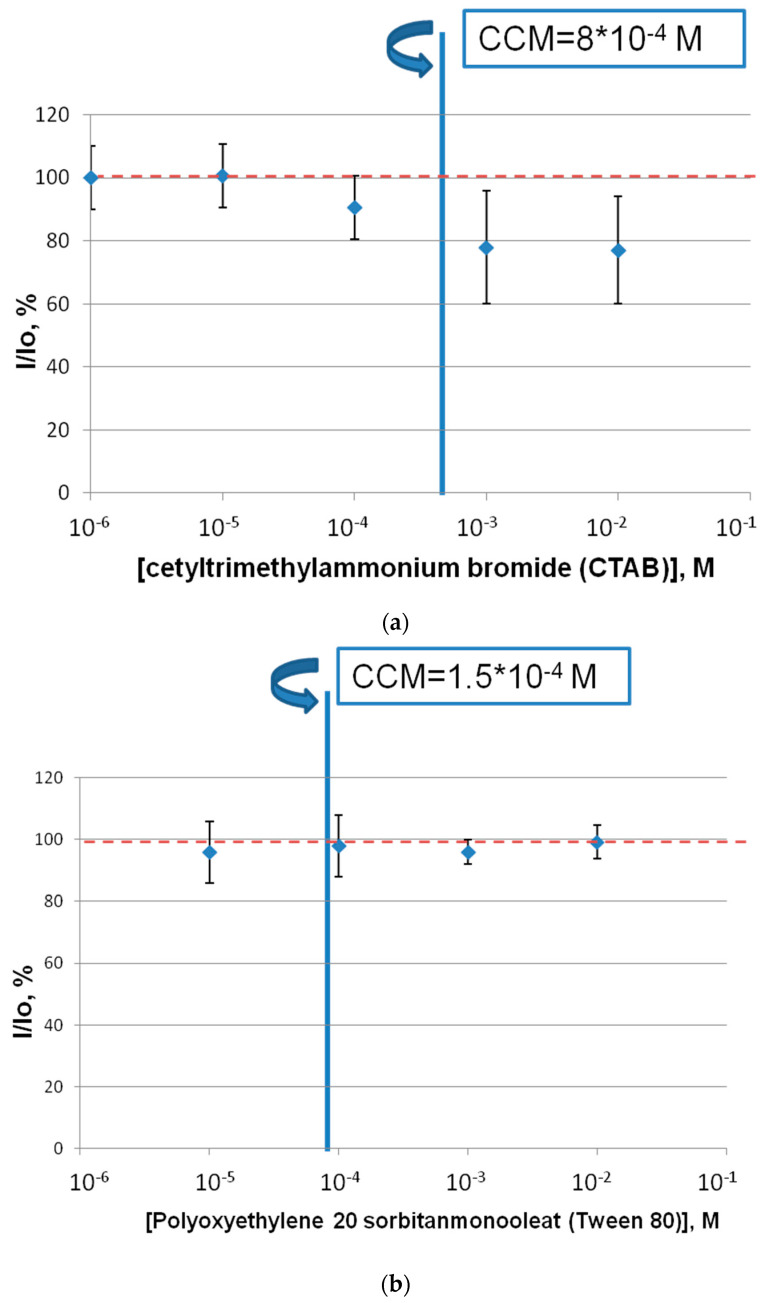
The residual luminescence intensity, I/I_0_, of the luminous bacterium *P. phosphoreum* in the presence of Tween 80 (**a**), CTAB (**b**) and SLS (**c**). The red dotted line means the values of light intensities of the luminous bacterium in the absence of the surfactants.

**Table 1 ijms-24-00515-t001:** The values of the parameters IC_50_ and EC_50_ for the Red + Luc enzyme system and luminous bacteria. The data are presented as the molar concentration (M). nd—not determined, low inhibition (%) at the maximum concentration tested.

Analyzed Surfactant	Red + Luc Enzyme System (IC_50_)	Luminous Bacterium Strain *P. phosphoreum* (EC_50_)
CTAB	nd	nd
SLS	10^−5^	10^−2^
Tween 80	nd	nd

**Table 2 ijms-24-00515-t002:** Toxicity data of the surfactants for aquatic species.

Surfactants	Species	End Point	Values
SLS	*Poecilia reticulata*	LC_50_	1.3 mg/L [37]
	*Ceriodaphnia affinis Lilljeborg*	LC_50_	12.6 mg/L [37]
	*P. phosphoreum*	EC_50_	70 mg/L [34]
CTAB	*Bacillus subtilis 6633*	LC_50_	1 × 10^−5^ mol/L [38]
	*P. phosphoreum*	EC_50_	0.5 mg/L [39]
Tween 80	*Nitrosomonas europaea*	EC_50_	1.51 mg/L [40]

**Table 3 ijms-24-00515-t003:** Characterization of the surfactants used in this study.

Surfactant	Abbreviations	CCM (10^−4^ M)	Structural Formula
Cetyltrimethylammonium bromide (cationic)	CTAB	8	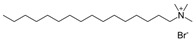
Polyoxyethylene 20 sorbitan monooleate (non-ionic)	Tween 80	1.5	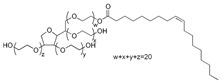
Sodium laurel sulfate (anionic)	SLS	82	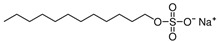

## Data Availability

Not applicable.

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
