# Peer review of "Toxicity of Different Types of Surfactants via Cellular and Enzymatic Assay Systems"

_ijms, 2022, doi:10.3390/ijms24010515_

Round 1

Reviewer 1 Report

This article can be publishable after several revisions. Specific comments are as follows.

1.      The daphnia magna LC50 values of these surfactants are suggested to be evaluated to further assess its toxicity.

2.      The advances should be introduced and improved. eg: Chinese Journal of Catalysis, 2022, 43, 2652–2664; Separation and Purification Technology, 2023, 304, 122401;

3.      The authors should also provide some discussion and on the reasons for the differences of their toxicity. 

3. There are some typographical and grammatical errors in the manuscript.

Reviewer 2 Report

Review of the manuscript entitled “Toxicity of Different Types of Surfactants via Cellular and Enzymatic Assay Systems” submitted to International Journal of Molecular Sciences, by Sutormin et al.

This manuscript aims to examine and compare the effects of three different types of surfactants (cationic, anionic and nonionic) on luminous marine bacteria and their coupled NAD(P)Н:FMN-oxidoreductase + luciferase (Red + Luc) enzyme system. The introduction and material and methods are well written, but the results and discussion sections are confusing, especially the discussion; in these two latter sections the English language must be improved; the discussion section needs to be almost fully rewritten, to organize the ideas. Below I give the authors several suggestions to improve the manuscript, especially the discussion. Moreover, several references are old and should be replaced by recent ones (from the last 10 years, preferably).

Abstract: the objective must be clearer. As far as I understood in the conclusion, you do not intend only to assess and compare the effects of three different types of surfactants but also to assess whether the Red + Luc enzyme system can be used as a more sensitive indicator of toxicity.

Introduction

Line 37 - not only

Line 45 – for clarity, you should be more specific regarding this range (0.0005-0.003 mg/L). Is this a range for EC50 values? If so, in what species and regarding which endpoint? The question is that the sentence, as it is, is vague. Since there are several reviews on detergents toxicity (e.g., https://doi.org/10.1007/s10311-014-0466-2, https://doi.org/10.1007/s11356-015-5803-x, among others) you should probably cite a review or any other paper that gives an idea about the toxicity of several surfactants.

Lines 47-48 – You mention that “The classification of surfactants also plays a crucial role in their toxicity level”. It is not surfactants’ classification that affects their toxicity, but their chemical structure.

Lines 54-56 – This statement would better be supported based on reviews, for instance those mentioned above (regarding my comment about line 45).

Also, and as mentioned in my comment above, the objective must be clearer.

Results

Lines 78-80 – In lines 76-78, you mention that CTAB did not cause any impact on the on the activity of the Red + Luc enzyme system. In lines 78-80 you mention that “In the same manner as with CTAB, Tween 80 concentrations, which were lower than the CCM concentration, influenced the activity of the coupled enzyme system”. You probably meant “did not influence”.

Moreover, note that referring “any impact” (or similar expressions, such as “inhibited” in line 82 or “not differ significantly from the control” in figure 1(b)) should be based on statistical analysis. Thus, you should have determined the concentrations significantly differing from control (e.g., using a one-way ANOVA).

Moreover, why don’t you describe the results regarding the toxicity of Tween 80 concentrations above the CCM?

Figure 1 – in the caption you refer “TWEEN 80”, despite you use “Tween 80” throughout the manuscript. The nomenclature should be uniformized. The same applies in Figure 2.

In the part (a) of the figure, in the horizontal axis, I suggest to add (CTAB) in parenthesis, to be concordant with the horizontal axis of parts (b) and (c). The same applies in Figure 2.

Also, exponential values are not commonly represented with commas (as shown in the horizontal axis of all parts of this figure). I advise to delete the comma. The same applies in Figure 2.

Part (b) – I suggest “not differing significantly from the control” or “no significant differences relative to the control” or “did not differ significantly from the control”. The current expression is not grammarly correct.

Part (c) - you probably meant “Concentrations of SLS found in aquatic ecosystems” or similar. Also, you should center the horizontal axis name, as in parts (a) and (b).

Lines 90-94 – this should be moved to the Discussion section, except the last sentence.

Table 1 – the caption of this table would benefit from being rewritten. For instance, there is no need to state “were determined during the toxicity assessment of the surfactants.” in the caption. Also, why wasn’t the IC50 value for Tween 80 determined? Based on Figure 1b, the two highest concentrations caused inhibition above 50%, which would allow to determine the IC50 value. Moreover, how did you determine the EC50 value for CTAB if inhibition was so low (Figure 2a, lines 102-104)?

Also, since “Photobacterium phosphoreum” was already mentioned in the introduction (line 55), the short name should be used after (P. phosphoreum). The same applies throughout the text (after line 55), such as in lines 99, 102, 106, 109, 115…

Moreover, why do you present the IC50 values with just 1 significant figure? Can you present values with higher precision?

Line 99 – In the Results section you show the results regarding the activity of the Red + Luc enzyme system first, and then the results of the bioluminescence of the bacteria Photobacterium phosphoreum, but you follow the opposite order in the Material and Methods section. The text becomes easier to read and follow if the same order is used in all sections.

Lines 103-104 – the expression “marine luminous bacteria did not show any valuable sensitivity to the presence of CTAB and Tween 80 samples.” should be improved, based on statistical analysis.

Figure 2 – what does the red dashed line means? This should be clarified in the caption.

Discussion

Lines 120-124 – you should provide one or two references to support this sentence. Moreover, this paragraph should be rewritten to make it clearer, and bibliographic references should be provided to support the ideas presented.

Lines 146-149 – Indeed, SLS concentrations from 0.5 to 4 mg/L, inhibited the activity of the Red + Luc enzyme system, which is classified as toxic effects. However, apparently (visually), there were not significant effects when the bacteria were exposed to this chemical (Figure 2). So, what is the ecological relevance of this result? I think you should mention that such toxicity will not be observed under natural conditions (based on Figure 2), which is due to the fact that enzymes are not exposed to the surfactants as they were in the experimental conditions of the present study.

Lines 150-151 – you refer “Intriguingly, SLS at the same concentration did not lead to any inhibition effects on the luminous bacteria.” Is that unexpected? Can you provide any explanation?

Lines 151-152 – the sentence “The SLS impact on the bioassay was found to depend on the type of the test organism involved [30].” must be rewritten

The discussion is confusing and very difficult to follow, and lacks comparison to results from previous studies. I suggest the authors to: a) compare the sensitivity of both bioassays and relate them to results of previous studies; b) compare the surfactants toxicity and relate it to results of previous studies (adding a summary table with EC50 values for some bacterial species exposed to these surfactants would help); c) discuss the importance of analyzing the activity of the coupled Red + Luc enzyme system; and d) discuss the ecological relevance of these findings – will marine bacteria be affected by the environmental concentrations? What will be the consequences?

Materials and Methods

Table 2 – Instead of “Surfactant type” maybe you mean just “surfactant”. You may add the surfactant type in this table, which is cationic, nonionic and anionic.

Line 225 – topic “4.2 Luminous Bacteria Assay” – how was the luminescence intensity measured, i.e, what equipment or technique?

Line 242-243 – you mention “I), 50 µL of the control solution was placed into 50 µL of the investigated surfactant solution”. This is confusing and should be clarified. Do you mean that 50 µL of distilled water were used instead of 50 µL of the surfactant solution, or do you mean that 50 µL of distilled water were added to 50 µL of the surfactant solution? The same confusion occurs in lines 274-275.

Lines 249-251 – what was the criterion for selecting these concentrations range?

Lines-251-252 – which method was used to determine the EC50 value? The same applies in lines 282-284.

Line 285 – topic “Data processing” – as mentioned above, in the topic Results (Lines 78-80), you should have determined the concentrations significantly differing from control. The information regarding such statistical analysis should be added here.

Line 153 – “Poecilia reticulata Peters” is not correct. The species is Poecilia reticulata, if you wish to add the name of the person who first recorded it, the name should not be in italics, and the year should also be added, e.g. Poecilia reticulata (Peters, 1859). However, given that the literature is not consensual regarding the year, I suggest to keep just “Poecilia reticulata”. The same happens in line 156.

Also, you mention that guppy fish has “greater sensitivity”. Are you comparing guppy’s sensitivity to that of P. phosphoreum? This is not clear.

Lines 152 – 160 – I think you should compare the sensitivity of P. phosphoreum to the sensitivity of other bacteria, rather than fish and cladocerans. Are your results similar to those obtained in previous studies?

Line 159 – You mention “membrane trophic activity”. I never heard of this expression associated to surfactants. I tried to find the reference you cite, but I could not find it online. Moreover, this is an old book (year 2005). I suggest to: a) explain what the expression means, how such activity occurs; and b), if possible, cite a reference from the last 10 years (maximum).

Conclusions

Lines 297-299 – what do you mean with the sentence “This is likely to be indicative of cooperative interactions between the proteins of the coupled Red + Luc enzyme system and the anionic surfactant.”?

Line 299  - what does it mean “cellular inferior”?

Lines 302-305 – do you think that it is so relevant to consider surfactant concentrations that affect the proteins if the organisms, when exposed to those concentrations do not exhibit significant effects? Maybe you could suggest bioassays with a longer exposure period to be able to assess if such concentrations could affect the organisms. Maybe you could suggest that these chemicals may have more serious effects if the cellular membrane was damaged (under such conditions, the chemicals could more easily affect the intracellular proteins).

The conclusion is confusing, long and lacks numerical data (for instance, you can add the IC50 and or EC50 values). The exception is the last paragraph, which is well written and clearly summarizes part of the work, highlighting the importance of some of the present results.
